# Impact of Agro-Byproduct Supplementation on Carcass Traits and Meat Quality of Hair Sheep and Wool × Hair Crossbreds Grazing on Fescue Pasture

**DOI:** 10.3390/ani14081217

**Published:** 2024-04-18

**Authors:** Jung Hoon Lee, Stephan Wildeus, Dahlia O’Brien, Brou Kouakou

**Affiliations:** 1Georgia Small Ruminant Research and Extension Center, Fort Valley State University, Fort Valley, GA 31030, USA; kouakoub@fvsu.edu; 2Agricultural Research Station, Virginia State University, Petersburg, VA 23806, USA; swildeus@vsu.edu (S.W.); dobrien@vsu.edu (D.O.)

**Keywords:** hair sheep, crossbreeding, grazing, soy hull, corn gluten feed, carcass trait, meat quality

## Abstract

**Simple Summary:**

This study investigated the impact of terminal sire (Dorset) use and supplementation with agro-byproducts, specifically corn gluten feed and soy hull, on the carcasses and meat quality of landrace hair sheep (Barbados Blackbelly and St. Croix) under rotational grazing. The crossbred lambs had greater carcass weights and more substantial wholesale cuts from the shoulder, rack, loin, and leg than their purebred counterparts. The purebred supplemented lambs showed comparable enhancements in their carcass characteristics to those of the crossbred lambs. Fresh lamb from the purebred animals displayed a more vivid red and yellow hue compared to the crossbred lambs. The meat from the purebred lambs was found to be softer than that from their crossbred counterparts. Supplementation played a more crucial role than crossbreeding in enhancing carcass quality under the conditions of these experiments. The protein content was notably higher in the meat from the crossbred lambs, while the supplemented lambs produced meat with an increased fat content. Pasture-only lambs had elevated levels of omega-3 and omega-6 fatty acids, displaying a healthier fatty acid profile than those receiving supplementary agro-byproducts. However, they benefited from the improved lipid oxidation stability and texture properties of the meat. The supplement type had no effects on the carcass characteristics, but it affected the meat composition.

**Abstract:**

The effects of breed type (purebred or crossbred) and supplementation of agro-byproducts on the growth, carcass characteristics, and meat quality of landrace hair (Barbados Blackbelly; BB and St. Croix; SX) lambs was evaluated. Thirty-six 7.5-month-old purebred hair (BB and SX; body weight = 24.1 ± 4.26 kg) and terminal sire (Dorset; DO) crossbred lambs (DO × BB and DO × SX; body weight = 31.4 ± 3.50 kg) rotationally grazed predominantly on Jesup tall fescue pasture during spring with or without agro-byproduct supplementation (soyhull or corn gluten feed at 2% of BW). Following a 77d grazing period, the lambs were harvested, and their carcass characteristics and meat composition were evaluated. Both supplementation and crossbreeding significantly increased their carcass weight and primal cuts, whereas only supplementation increased (*p* < 0.01) the dressing percentage, and crossbreeding increased the shear force (*p* < 0.01). Regardless of breed type, supplementing agro-byproducts improved the lipid oxidation stability and texture properties of the fresh lamb, whereas the pasture-only lambs had healthier fatty acid profiles compared with the supplemented lambs. The results indicate that both terminal sire crossbreeding and byproduct supplementation can be used to affect the carcass characteristics and meat composition of landrace hair sheep lambs.

## 1. Introduction

Red meat, recognized for its significant contribution to human nutrition by supplying high-quality protein, iron, zinc, and vitamin B complex, faces consumer scrutiny due to its high saturated fat content, which is linked to health concerns [1,2]. In recent decades, a shift in consumer preferences towards healthier red meat options has emerged, spotlighting grass-fed and naturally raised products as preferable alternatives [2,3,4]. This trend is not only a reflection of growing health consciousness but also a nod towards environmental and ethical considerations in meat production. The United States (US) sheep industry, as highlighted in a comprehensive review by the National Research Council, has seen a steady climb in demand for high-quality, forage-finished lamb, especially among ethnic and health-conscious consumer groups [5,6]. This shift presents a notable opportunity for the US sheep industry expansion, particularly through the utilization of alternative breeds like hair sheep. Hair sheep breeds, such as the Barbados Blackbelly (BB) and St. Croix (SX), are prized for their low maintenance, demonstrating resilience without the need for intensive management practices like deworming and assistance during lambing [7,8]. Despite their appeal, these breeds typically yield smaller carcasses (<50 kg vs. >50 kg) and slower growth rates (<200 g/d vs. >200 g/d) compared to traditional wool breeds in grazing and feed lot trials [7,8,9,10]. Crossbreeding strategies employing wool breeds, such as Dorset and Suffolk rams, have been explored as a means to enhance the growth performance of hair sheep, promising heavier carcasses and more desirable cuts for the consumer market [11,12].

Previous research has demonstrated the potential of crossbreeding to align with market demands for quality lamb meat [13,14,15]. However, achieving optimal carcass weight and quality often necessitates nutritional interventions beyond what forage-based systems can offer, particularly due to the limitations posed by seasonal forage availability and quality [16,17]. The incorporation of legumes and dietary supplements into pasture systems has been identified as a strategy to bolster animal performance and carcass quality. Traditional grain-based supplements (corn, grain sorghum, cottonseed meal, and soybean meal), while effective, may not be economically viable for growing grazing ruminants and can disrupt ruminal pH and fiber digestion [18,19]. Agro-byproducts like soy hull (SH), containing about 73% fiber and 9.4% protein [19], and corn gluten feed (CGF), containing nearly 18% fiber and 22% protein [20], emerge as cost-effective alternatives, offering balanced nutritional profiles that support production goals without the drawbacks associated with conventional grains. However, the impacts of supplementing SH or CGF on animal performances and carcass qualities in the grass-based hair sheep production system are not well known because different forms and energy levels of supplementations influence carcass composition and meat properties in meat-producing animals [21,22,23]. The aim of this study was to evaluate the effects of terminal sire crossbreeding and dietary supplementation on the carcass traits and meat quality of hair sheep grazing on fescue pastures.

## 2. Materials and Methods

### 2.1. Animal Management and Experimental Design

This grazing study was approved by the Committee on the Care and Use of Agricultural Animals at Virginia State University (Approval Number: A3800-01). The research was carried out at the university’s small ruminant facility located in Petersburg, VA, USA. The animals used in this study were managed with strict adherence to the university’s Agricultural Animal Care and Use Guide. The research flock was maintained on a pasture using rotational grazing and was managed under an 8 mo accelerated mating system. The experiment was part of a project that evaluated the use of a terminal sire to improve the productivity of a landrace hair (BB and SX). Dorset (DO) rams were selected as the terminal sire breed to accommodate the need for extended seasonal breeding under accelerated mating and a moderately sized breed to avoid lambing difficulties in ewes. Ewes were randomly allocated to be mated by like-breed sires or DO rams in 9 single-sire mating groups (3 BB, 3 SX, and 3 DO ram groups). The lambs used in the study were born on the pasture in August, weaned at 9 weeks of age, and moved back on the pasture after a 2 mo transition in a dry lot.

For the study, 36 ram lambs, equally representing purebred (BB; BW = 23.6 ± 4.4 kg, and SX; BW = 24.6 ± 4.2 kg) and crossbred breed types (DO × BB; BW = 30.6 ± 3.8 kg and DO × SX; BW = 31.4 ± 2.9 kg), were allocated at random to a pasture-only group or to pasture groups supplemented with either SH or CGF at 2% BW (Table 1). The lambs grazed as single group in a rotational grazing system on Jesup tall fescue pastures that contained the Max-Q^©^ (Pennington Seed Co., Williamette Valley, OR, USA), endophyte. The lambs were moved following visual appraisal of biomass in 3 to 5 d intervals. Pasture samples were collected as the animals were moved to a new section and pooled for the duration of the study to estimate nutritional quality (Table 1). Supplement was provided to the individual lambs using Calan^©^ gate feeders (American Calan, Northwood, NH, USA), providing the daily allowance for each lamb. The study lasted for a period of 77 d from March to June. Body weight was recorded at the beginning and end of the trial and in 14 d intervals to adjust the feeding levels of the supplements.

### 2.2. Carcass Processing and Evaluation

Upon the completion of the 77-day grazing period, the lambs were transported to a USDA-certified slaughterhouse at Fort Valley State University, located in Fort Valley, GA, USA, where they were processed according to established protocols. The hot carcass weight (HCW) of each lamb was recorded immediately post-slaughter, and the carcasses were then cooled at 2 °C for 24 h. Following this, the cold carcass weight (CCW), carcass shrinkage, and dressing percentage (DP)—the percentage of live weight that translates to carcass weight—were assessed. The ultimate pH of the muscle was determined by measuring it between the 12th and 13th ribs at 24 h postmortem using a portable pH meter equipped with a penetrating probe (Pakton^®^ Model OKPH1000N, Fisher Scientific, Pittsburgh, PA, USA). On the 2nd day following slaughter, the carcasses were dissected into primal cuts, and the weight of each cut was recorded. Intramuscular (*Longissimus* muscle; LM) fat depot from the fifth through seventh ribs was excised from each carcass, and the subcutaneous fat was also removed for fatty acid analysis. The loin from each carcass was sliced into 2.5-cm loin chops and then used to measure the fresh meat color (CIE L* a* b*), Warner–Bratzler shear force (WBSF) values, and cooking losses.

The assessment of the fresh meat color involved the utilization of the CIE L* a* b* coordinates, measured on the surface of 4 loin chops after a 45 min rest at 4 °C using a HunterLab Color instrument (Minolta Chromameter, Model CR-200, Minolta, Japan) using illuminant D65 as a light source. This was followed by cooking loss evaluations and Warner–Bratzler shear force (WBSF) analyses of the cooked meat to assess its tenderness according to the procedures described by Lee et al. [24]. The differences in the weight of the chops before and after cooking was reported as the percentage cooking loss. Two cores were taken from each cooked chop, and the WBSF values were assessed using a TA-XT2 texture analyzer fitted with a Warner–Bratzler shear attachment (Texture Technologies Corp., Scarsdale, NY, USA). The myoglobin (Mb) and metmyoglobin (MetMb) contents were determined following Krzywicki’s methodology [25] using the ground LM (5.0 g) via spectrophotometric analysis. Both contents were measured at 525, 572, and 700 nm using a Shimadzu (model UV-2401 PC) spectrophotometer (Shimadzu Scientific Instruments Inc., Columbia, MD, USA), and the concentration of Mb (mg/g muscle) and percent MetMb (%) were calculated [25]. Similarly, the degree of lipid oxidation was quantified using the thiobarbituric acid reactive substances (TBARS) method using ground LM (5.0 g), as described by Buege and Aust [26], using 1,1,3,3-tetramethoxypropane (TMP) for the preparation of a standard curve of malondialdehyde (MDA) and expressing the results in terms of MDA concentration per kilogram of sample.

### 2.3. Meat Composition

The nutritional makeup of the LM samples was established using methods recommended by the AOAC [27]. The total lipids were extracted from 3.0 g of LM or 0.1 g of subcutaneous fat samples using chloroform/methanol (2:1 *v*/*v*) according to the procedures described by Lee et al. [24]. The extracted lipids were saponified and esterified according to the AOCS method [28] for preparation of fatty acid methyl esters (FAMEs). The prepared FAMEs were analyzed using a Thermo Electronic (Austin, TX, USA) gas chromatography apparatus (Model TRACE GC Ultra) equipped with an automatic sampler Model AS-3000 (Thermo Electronic Co., Waltham, MA, USA). A 0.25 mm i.d. by 60 m long fused silica SP-2380 capillary column (Supelco, Inc., Bellefonte, PA, USA) was used to separate the methyl esters, which were detected using a flame ionization detector (FID). The injection temperature was 240 °C, and the column temperature was programmed from 130 °C to 220 °C at 4 °C/min. Helium was the carrier gas, with a flow rate at 1.6 mL/min and a split ratio of 30:1. Furthermore, the identification and quantitation of individual FAMEs in the sample were also completed according to the AOCS method [28].

### 2.4. Statistical Analysis

The data from this study were evaluated using a completely randomized design framework, incorporating a factorial arrangement for the treatments. This analysis was performed using the SAS Institute’s MIXED procedure (Cary, NC, USA), treating each animal as an individual experimental unit. The impact of breed (purebred and crossbred) and dietary supplementation (pasture alone, soy hull, and corn gluten feed) was examined, along with their interactive effects. The significance of the differences between the means was determined using the least squares means method, applying the PDIFF option in SAS for assessing both the main and interactive effects, with the significance set at *p* < 0.05 and trends noted at *p* < 0.1.

## 3. Results and Discussion

### 3.1. Body Weight Gains and Carcass Traits

The average daily gain (ADG) was notably influenced (*p* < 0.0001) by breed type, the presence of dietary supplements, and the interactions (*p* < 0.05) between the two (Table 2). The lambs receiving either supplement showed a substantial increase in ADG, nearly double that of the pasture-only lambs. However, the supplement type had no effect on ADG and carcass weight. The crossbred lambs exhibited a 1.4-fold increase (*p* < 0.001) in ADG compared to the purebred hair sheep. The highest ADG was recorded in the crossbred lambs fed with CGF or SH (234.47 or 249.60 ± 9.516 g), which was significantly higher (*p* < 0.05) than those grazing without supplements, irrespective of breed type (purebred and crossbred; 106.92 and 129.52 ± 9.516 g). The supplemented purebred lambs showed a moderate increase in ADG (CGF or SH; 176.49 or 167.06 ± 9.516 g), positioned between the two extremes (*p* < 0.05). The FBW also varied significantly (*p* < 0.0001) with breed and supplementation, favoring the crossbred and supplemented lambs over the purebreds and those grazing only (Table 2). However, the interaction between breed type and supplementation did not significantly affect the FBW (*p* = 0.1458).

This research confirmed the expected impact of breed type and dietary supplementation on lamb growth. Typically, landrace hair sheep are smaller with slower growth rates compared to wool breeds in the US [29]. Consequently, the sheep resulting from terminal sire crossbreeding (wool × hair) should have increased sizes and growth rates. This increase was quantified at approximately 40% for the landrace hair sheep lambs on the pasture in this study using a moderate size terminal sire breed. The feeding regime also has a major influence on animal weight gain because of more easily utilized dietary protein. In this study, supplementation at 2% BW nearly doubled the growth rate of the rotationally grazed lambs. The benefit of supplementation was most pronounced in the crossbred lambs, allowing them to fully express their growth potential. No effects on growth rate were derived from the higher CP content in the diet of the CGF-supplemented lambs, though research has suggested that increases in dietary protein can improve the growth performance of pasture-fed lambs [17]. The interaction between breed and supplementation contributed significantly to the variations observed in the ADG.

Differences (*p* < 0.0001) in the hot and cold carcass weights and loin area were observed, showing higher values for the crossbred and supplemented lambs compared to the purebred and pasture-only lambs, though with some variability (Table 2). No significant interaction was found between the breed type and supplementation for HCW and CCW (*p* = 0.0602 and *p* = 0.1084, respectively). The carcass shrinkage (3.89 to 6.04 ± 1.431%) remained consistent across the treatments, while the dressing percentage was lower (*p* < 0.001) in the pasture-only lambs compared to those receiving supplements. This can be attributed to the increase in digestive tract size and decrease in external fat cover that are typical of forage diets, impacting the dressing percentage negatively. In ruminants, forage diets generally increase digestive tract size and decrease external fat cover, resulting in lower dressing percentages in the carcasses [30]. McClure et al. [31] also reported that lambs that grazed on pastures had heavier intestinal weights and thinner layers of external fat cover than those fed concentrates. Because of the larger and heavier carcasses obtained from both the crossbred and supplemented lambs, both breed type and supplementation might affect loin area. Interestingly, only the breed type significantly influenced the loin area, with the crossbred lambs showing larger (*p* < 0.05) loin areas than their purebred counterparts.

Significant differences in the primal cuts of the carcasses were observed (*p* < 0.01) for breed type and diet, with the crossbred and supplemented lambs yielding heavier (*p* < 0.001) cuts (shoulder, fore-shank, rack, breast, loin, leg, and hind-shank) across most categories except for the neck. Specifically, the rack cuts demonstrated a notable interaction (*p* < 0.05) between breed type and supplementation, indicating a complex relationship between these factors in influencing carcass composition. Within either the CGF or SH supplemented groups, the crossbred lambs had heavier weights of rack cuts (2.24 or 2.41 vs. 1.98 or 1.51 ± 0.121 kg,) than the purebred lambs supplemented with SH, but no differences were found in the weights of the rack cuts (1.23 and 1.60 ± 0.121 kg) from the two different breed types in the pasture-only lambs and purebred lambs supplemented with SH. Furthermore, the supplemented crossbred lambs had the heaviest weights of rack cuts, and the purebred pasture-only lambs had the lightest.

### 3.2. Quality Characteristics of Lamb Chops

This study also assessed the quality traits of the lamb chops, noting the influence of breed (purebred and crossbred), diet (pasture-only, soy hull, and corn gluten feed), and their interaction on the CIE L* (lightness), a* (redness), and b* (yellowness) color metrics (Table 3). The CIE L*, a*, and b* values of the lamb chops were affected by supplementation, breed type, and their interaction. A notable (*p* < 0.01) two-way interaction (breed type × supplementation) was observed in the CIE L*, a*, and b* values. This contrasts with earlier studies that found that breed variations among native sheep breeds used for meat production did not significantly alter the CIE L*, a*, and b* color coordinates of the fresh lamb [32]. The loin chops from the pasture-only lambs exhibited markedly higher CIE L* and b* values compared to those from the lambs supplemented with CGF and SH. In contrast to the observations made by Suliman et al. [32], our investigation revealed that the breed type and diet significantly impacted the color metrics of the lamb chops, with notable interaction effects between these variables. Specifically, the chops derived from purebred lambs showcased higher degrees of redness and yellowness (elevated CIE a* and b* values) compared to those obtained from the crossbred lambs. Similarly, the loin chops from the pasture-only lambs demonstrated greater lightness and yellowness (higher CIE L* and b* values) than those from the supplemented lambs. An interaction between breed type and diet was observed, indicating that the pasture-only crossbred lambs possessed the most pronounced CIE L* values (*p* < 0.0001), while the crossbred lambs supplemented with SH had the least-pronounced values (38.75 and 35.61 ± 0.356, respectively). Intermediate CIE L* values were noted in the pasture-only and SH-supplemented purebred lambs, as well as in the CGF-supplemented crossbred lambs (37.02, 37.36, and 37.27 ± 0.356, respectively). Regarding the CIE a* values of the chops, the CGF-supplemented crossbred lambs showed higher redness values (*p* < 0.01) compared to the pasture-only or SH-supplemented lambs (12.24 vs. 11.34 and 11.36 ± 0.199, respectively), yet no significant variance was observed among the purebred groups across the different diets (pasture-only, CGF-supplemented, and SH-supplemented; 12.96, 12.63, 12.71 ± 0.199, respectively). Independent of diet, the purebred lambs exhibited superior (*p* < 0.01) CIE a* values for their chops compared to those from the pasture-only and SH-supplemented crossbred lambs but were similar to the CGF-supplemented crossbred lambs. The highest CIE b* values were noted in the chops from the pasture-only and SH-supplemented purebred lambs (*p* < 0.0001), whereas the SH-supplemented crossbred lambs had the lowest value (11.41 and 9.91 ± 0.189, respectively), with the pasture-only and CGF-supplemented crossbred lambs, along with the CGF-supplemented purebred lambs, showing intermediate CIE b* values (11.08, 10.97, and 10.54 ± 0.189, respectively).

This variance highlights the influence of both genetic makeup and environmental conditions, including dietary factors, on the quality and visual attributes of the meat—aspects that are pivotal for consumer assessments related to color, fat marbling, and moisture retention capabilities. Previous studies have corroborated that the interplay between genetics and environment plays a significant role in shaping the meat production qualities of ruminants, with the meat color being particularly affected by the myoglobin (Mb) concentration as well as the moisture and fat content within the consumable muscle tissue [33,34]. In our investigation, the genetic background of the lambs notably influenced the CIE a* (redness) and b* (yellowness) color metrics of the lamb chops. Nonetheless, the variations observed in the CIE color metrics could not be conclusively attributed to differences, given the uniform Mb levels and proximate composition of the LM muscle across the studied breeds.

The lambs on the pasture-only diet displayed a higher (*p* < 0.0001) moisture content and a reduced (*p* < 0.0001) fat content in their LM muscle, in contrast to those supplemented, though no significant differences were observed in the Mb content among the three different dietary groups (Table 3). The increased moisture content in the LM muscle of the pasture-only lambs could impede the absorption of visible light, potentially enhancing the reflectance on the surface of the lamb chops, thus explaining the higher CIE L* values observed in this study. Moreover, the loin chops from the supplemented lambs demonstrated a significantly lower yellowness (CIE b*), attributed to the elevated (*p* < 0.0001) fat content in the LM muscle of these lambs. This finding aligns with Lee et al. [35], who observed that small ruminants on a hay diet presented a more yellowish hue (higher CIE b* value) in their LM muscle compared to those fed a concentrate-rich diet.

Despite the observed variances in the color metrics, the myoglobin (Mb) concentration in the *Longissimus* muscle (LM) exhibited no significant differences when comparing the breeds or dietary regimens, pointing towards the influence of additional factors on meat coloration. The observed mean Mb levels in the LM muscle spanned from 5.78 to 9.52 mg/g, with the average Mb concentration in major Australian lamb meat types reported as approximately 6.64 mg/g of muscle [36]. Typically, the Mb content is influenced by genetic factors such as species, age, sex, and muscle function, with muscles engaged in more rigorous activity tending to contain higher Mb levels [37]. Interestingly, the study noted a significant interaction between breed type and dietary supplement in relation to the percentages of metmyoglobin (MetMb) within the LM, albeit without a direct significant impact from either breed type or supplementation alone. The LM muscle from the SH-supplemented purebred and CGF-supplemented crossbred lambs demonstrated elevated (*p* < 0.0001) MetMb percentages (41.35 and 39.26%, respectively) compared to the SH-supplemented crossbreds (23.56 ± 3.220%). Meanwhile, the pasture-fed crossbreds and CGF-supplemented purebreds displayed intermediate levels of MetMb (37.72 and 37.67 ± 3.220%, respectively), significantly differing from the SH-supplemented crossbreds. The coloration of fresh red meat predominantly relies on Mb concentration and its oxidation state, cycling through deoxymyoglobin (DeoxyMb), oxymyoglobin (OxyMb), and metmyoglobin (MetMb) forms, with the balance between these states being largely dictated by the oxygen saturation levels of Mb [38]. MetMb formation typically occurs within the meat’s interior due to oxygen scarcity below the surface layer [39], leading to Mb oxidation into MetMb, which is closely linked with meat discoloration phenomena.

This study further explored how dietary supplements impact the levels of thiobarbituric acid reactive substances (TBARS), which serve as indicators for the extent of lipid oxidation within the meat. The findings revealed that the lambs fed solely on the pasture and those supplemented with CGF exhibited higher (*p* < 0.03) TBARS values, indicating greater lipid oxidation, compared to the SH-supplemented lambs (Table 3). TBARS measurement is a widely accepted method for assessing the degree of lipid oxidation in meat products, a process that can lead to the formation of free radicals. These radicals are capable of oxidizing meat pigments and producing undesirable rancid odors and flavors [40]. In this study, the variations in lipid oxidation rates among the meat samples were solely attributed to diet. Surprisingly, the SH-supplemented lambs displayed lower (*p* < 0.05) TBARS values in their loin chops, despite there being minimal differences in the fatty acid composition of the *Longissimus* (LM) muscle across the three diet groups, including unsaturated fatty acids. Moreover, the results demonstrated that the ultimate pH levels in the muscle, ranging from 5.58 to 5.69 ± 0.061, were consistent across all the dietary treatments, hinting that factors other than diet, occurring post-mortem, might have a pivotal influence on meat pH levels. Typically, the muscle pH of fresh lamb post-mortem decreases, stabilizing between 5.3 and 5.8 within 24 h [41]. It is recognized that several variables, including breed, age, diet, and stress levels, can impact the final pH value of red meat [42].

The cooking losses and Warner–Bratzler shear force (WBSF) values, indicators of meat tenderness, also varied with the breeds and diets, illustrating the intricate effects of genetics and diet on meat characteristics. This study demonstrated that the cooking losses of loin chops from the experimental groups were not influenced by the treatments applied (Table 3). It has been previously established that factors such as muscle pH, the proximate composition of the meat, aging duration, and cooking temperature can all impact cooking losses [43,44]. Additionally, the water holding capacity (WHC) and the amount of intramuscular fat are known to affect cooking loss in meat [45,46]. In this study, the absence of differences in cooking losses across the breed types and diets could be attributed to the consistent ultimate muscle pH levels observed in the fresh lamb (Table 3), despite the supplemented lambs having a higher intermuscular fat content and lower moisture compared to those fed the pasture only. The tenderness of the cooked loin chops, as measured by the WBSF values, was significantly affected (*p* < 0.05) by both the breed type and the dietary supplement, as well as their interaction. The range of shear force values detected in this study (2.87 to 4.34 kg/cm^3^) exceeded the tenderness threshold accepted by consumers in Australia and New Zealand (3 kg/cm^3^) [47]. Specifically, the loin chops from the crossbred lambs exhibited significantly higher (*p* < 0.0001) WBSF values compared to those from the purebreds. The loin chops from the pasture-only lambs had higher (*p* < 0.05) WBSF values compared to those given CGF, but these values did not significantly differ from the lambs supplemented with SH. No marked differences were noted in the WBSF values between the two supplementation groups. These findings align with those of Shackelford et al. [48], who noted the influence of breed on the tenderness of fresh lamb meat. It has been commonly observed that lambs on concentrated diets tend to produce more tender meat compared to those that graze only [49,50]. An interaction between breed type and supplementation was evident in the WBSF values of the cooked chops from the experimental lambs. The crossbred lambs, whether fed the pasture only or supplemented with SH, had higher (*p* < 0.008) WBSF values than the purebred lambs supplemented with either SH or CGF (3.72 or 4.34 vs. 2.87 or 2.93 ± 0.479 kg/cm^3^, respectively), although the differences between the former groups were not statistically significant. Moreover, the pasture-only purebred and crossbred lambs had intermediate WBSF values (3.20 and 3.72 ± 0.479 kg/cm^3^, respectively), which were lower than those from the crossbred lambs supplemented with SH.

### 3.3. Chemical Composition of Longissimus Muscle (LM) and Fat Depots

The proximate composition analysis revealed that the moisture content in the LM was significantly affected by diet but not breed type (Table 4). The pasture-only lambs had higher (*p* < 0.0001) moisture percentages in the LM, ranging from 74.67 to 76.41%, in contrast to those that were supplemented. Breed type significantly (*p* < 0.01) influenced the protein levels in the LM, but not diet. The LM of the crossbred animals exhibited higher (*p* < 0.01) protein concentrations, between 20.83 and 21.70%, compared to that of the purebred lambs. The supplemented lambs had higher (*p* < 0.0001) fat levels than the pasture-only lambs, with their fat content varying from 0.60 to 1.93%. Similarly, the ash content in the LM was influenced by diet but not breed type. The pasture-only lambs had higher (*p* < 0.01) ash levels than those supplemented with SH, with the CGF-supplemented lambs having intermediate ash levels. These findings emphasize the critical role of diet in modifying the nutritional makeup of the LM, where dietary supplements contributed to an increased fat level but reduced moisture content compared to a strictly pasture-only diet. The composition and utilization of nutrients play a vital role in muscle development, and the energy content of the diet may help explain the observed variations in the proximate composition of the LM. This aligns with the existing literature indicating that concentrate-rich diets tend to increase the fat content in muscle [51,52,53], highlighting the substantial impact of diet, alongside breed, age, and sex, on the proximate muscle composition in ruminants.

The link between red meat consumption and health risks is often attributed to its high saturated fat content, which is a result of the hydrogenation process of unsaturated fats in the rumen. The nutritional profile of lamb meat, particularly its fatty acid composition, is determined by a range of factors, both inherent (such as breed, gender, age, and slaughter weight) and external (including the type and amount of diet) [51,54]. Adjusting the diet appears to be the most viable method for altering the fatty acid makeup in lamb meat for enhanced health benefits. Our study identified various fatty acids in the lamb LM and subcutaneous fat, encompassing saturated (SFA), monounsaturated (MUFA), and polyunsaturated (PUFA) types. Specifically, the analysis highlighted the presence of seven SFAs (C10:0, C12:0, C14:0, C16:0 iso, C16:0, C17:0, C18:0), eight MUFAs (C14:1n5, C16:1n7, *trans*, C16:1n7, C17:1n7, C18:1n9, *trans*, C18:1n11, *trans*, C18:1n9, C20:1n9), and nine PUFAs (C18:2n6, *trans*, C18:2n6, C18:2c9,t11, C18:3n3, C18:3n6, C20:2n6, C20:4n6, C20:5n3, and C22:5n3). Breed type and diet were significant determinants of the fatty acid profiles, with no notable interaction effects observed on the fatty acid distribution, except for lauric (C12:0) acid in the LM.

This research delineates clear variations in the fatty acid composition of the intramuscular fat (LM) across breed types and as a result of supplementation. It was observed that the crossbred lambs had significantly higher (*p* < 0.05) levels of lauric (C12:0) and palmitic (C16:0) acids than the purebred lambs, with the impact of supplementation being more pronounced on medium and long-chain fatty acids (Table 4). Specifically, the medium (C10 to C12) and long-chain (C14 to C22) fatty acids in the LM were notably influenced by the type of supplementation. The pasture-only lambs had increased (*p* < 0.0001) levels of capric (C10:0) and lauric (C12:0) acids in their LM compared to the supplemented lambs. Furthermore, the pasture-only lambs had elevated (*p* < 0.001) proportions of various long-chain fatty acids, including myristoleic (C14:1n5), *trans*-7-hexadeanoic (C16:1n7, *trans*), palmitoleic (C16:1n7), margaric (C17:0), heptadecenoic (C17:1n7), vaccenic (C18:1n11, *trans*), linolelaidic (C18:2n6, *trans*), α-linolenic (C18:3n3), eicosenoic (C20:1n9), eicosadienoic (C20:2n6), arachidonic (C20:4n6), eicosapentaenoic (C20:5n3), and docosapentaenoic (C22:5n3) acids, compared to those receiving the supplements. However, lower levels (*p* < 0.001) of *iso*-palmitic (C16:0 iso), palmitic (C16:0), oleic (C18:1n9), and γ-linolenic (C18:3n6) acids were observed. An interaction (*p* < 0.001) was noted between the breed type and the concentration of C12:0 acid in the LM, with the crossbred lambs showing a difference compared to the pasture-only and supplemented lambs, while this was not the case for the purebred lambs. This study also demonstrated that palmitic (C16:0), stearic (C18:0), and oleic (C18:1n9) acids were the predominant fatty acids in the LM. The palmitic (C16:0) acid contents, linked to higher cholesterol levels and an increased risk of coronary heart diseases [55], were lower (*p* < 0.05) in the purebred and pasture-only lambs (Table 4). While previous research has suggested breed effects on lamb fatty acid profiles [56,57,58], the current study found no significant breed-related differences in fatty acid levels except for lauric (C12:0) and palmitic (C16:0) acids, possibly due to similar genetic factors affecting muscle growth and development across breeds. Generally, forage-fed ruminants produce meat with comparable saturated fat levels, less monounsaturated fat, and more polyunsaturated fat than their concentrate-fed counterparts [47]. Lambs grazing on pastures exhibit higher levels of omega-3 polyunsaturated fatty acids (PUFA) and conjugated linoleic acids (*CLA*) [59], attributed to forage diets promoting fibrolytic microorganism growth in the rumen, which in turn enhances the production of *CLA* precursors [60]. In this study, the LM from the pasture-only lambs contained less saturated (C16:0) and monounsaturated (C18:1n9) fats and more polyunsaturated fats (including omega-3 and -6 PUFAs) compared to those fed the agro-byproduct supplements, possibly due to the forage’s influence on ruminal biohydrogenation processes or its intrinsic fatty acid composition. However, the levels of CLA and C18:1n11, *trans* were not distinctly altered by the diet.

Within the subcutaneous fat, the concentration of *trans*-7-hexadecenoic acid (C16:1n7, trans) was distinctly influenced by breed type (Table 5). The crossbred lambs exhibited a higher (*p* < 0.05) concentration of C16:1n7, *trans* in their subcutaneous fate compared to the purebred lambs, while the pooled mean concentrations of the other fatty acids in the subcutaneous fat did not significantly vary (*p* > 0.1) between the two breed types. Typically, lean breed lambs are known to possess a higher ratio of polyunsaturated fatty acids (PUFA) and lower levels of saturated fatty acids (SFA) and monounsaturated fatty acids (MUFA) in comparison to heavier breed lambs [56,57,61]. Nevertheless, we did not observe notable differences in the fatty acid profiles of the subcutaneous fat between the purebred and crossbred lambs, with the exception of trans-7-hexadecenoic acid. Supplementation had a notable impact on the profile of individual fatty acids within the subcutaneous fats of the lambs, with long-chain fatty acids (C16 to C22) being particularly affected by diet. The pasture-only lambs exhibited elevated levels of specific fatty acids such as stearic (C18:0), vaccenic (C18:1n11, trans), rumenic (conjugated linoleic acid, CLA, C18:2c9,t11), g-linolenic (C18:3n6), eicosapentaenoic (C20:5n3), and docosapentaenoic (C22:5n3) acids in their subcutaneous fat compared to those supplemented with agricultural by-products. Conversely, lower levels of palmitic (C16:0), margaric (C17:0), heptadecenoic (C17:1n7), linoleic (C18:2n6), and a-linolenic (C18:3n3) acids were noted. However, no significant differences were observed in the concentrations of C16:0, C20:5n3, and C22:5n3 acids in the subcutaneous fat between the pasture-only lambs and those supplemented with SH; similarly, the levels of C17:0 and C17:1n7 acids did not vary between the pasture-only lambs and those supplemented with CGF. These findings are similar to those of Webb and O’Neill [59], who documented that pasture-grazed lambs exhibited higher levels of omega-3 PUFA and CLA in their fat stores compared to those receiving grain supplements.

## 4. Conclusions

Both agro-byproduct supplementation and the use of a terminal sire significantly impacted the daily gain, final weight, carcass characteristics, and fatty acid profiles. The supplementation had a greater impact on the growth performance, while the characteristics and meat composition were selectively impacted by both the breed type and supplementation. The supplement type did not affect the growth performance; however, it selectively affected some carcass characteristics and the fatty acid composition. The use of terminal sire mating and supplementation are strategies that can help to improve the overall productivity of a pasture-based production system while creating a product that addresses the expectations of health-conscious customers.

## Figures and Tables

**Table 1 animals-14-01217-t001:** Chemical composition of tall fescue, soyhull, and corn gluten feed.

Item	Tall Fescue	Soyhull	Corn Gluten Feed
Chemical composition, %DM			
Dry matter, DM	88.6	90.0	87.8
Ether extract, fat	2.10	1.68	2.85
Crude protein, CP	10.4	12.5	17.9
Ash	6.61	4.52	6.37
Acid detergent fiber, ADF	50.4	49.4	17.0
Total digestible nutrient, TDN	58.2	56.0	76.0
Net energy, Mcal/kg	0.66	0.75	1.32
Fatty acid, %			
C14:0	0.44	0.43	0.40
C16:0	21.02	14.09	10.95
C16:1n7	1.05	0.25	0.01
C18:0	2.86	4.98	3.18
C18:1n9	4.39	19.07	23.61
C18:2n6	14.21	48.05	54.95
C18:3n3	41.01	10.13	2.33
C18:3n3	49.71	7.69	2.33
∑SFA ^a^	22.22	19.26	14.53
∑MUFA ^b^	5.87	20.80	23.62
∑PUFA ^c^	63.27	55.34	57.28
∑n-3 ^d^	49.71	7.69	2.33
∑n-6 ^e^	13.56	47.65	54.95
∑n-6/n-3	0.27	6.20	23.58

^a^ SFA = saturated fatty acids; C14:0, C16:0, and C18:0. ^b^ MUFA = monounsaturated fatty acids; C16:1n7 and C18:1n9. ^c^ PUFA = polyunsaturated fatty acids; C18:2n6 and C18:3n3. ^d^ n-3 = n-3 PUFA; C18:3n3. ^e^ n-6 = n-6 PUFA; C18:3n6

**Table 2 animals-14-01217-t002:** Daily weight gain and carcass traits in purebred and crossbred (wool × hair) hair sheep lambs grazed on tall fescue pastures with or without supplementation (soy hull; SH or corn gluten feed; CGF).

	Treatment					*p*-Value		
	Breed Type, B		Diet, D			Main		Interaction
Item	Purebred	Crossbred	Pasture Only	Pasture + CGF	Pasture + SH	B	D	B × D
Average daily gain (ADG), g	150.2 ^a^	204.5 ^b^	118.2 ^a^	208.3 ^b^	205.5 ^b^	0.0001	0.0001	0.0132
Fasting body weight (FBW), kg	31.59 ^a^	42.26 ^b^	32.19 ^a^	40.03 ^b^	38.56 ^b^	0.0001	0.0001	0.1458
Hot carcass weight, kg	13.75 ^a^	18.50 ^b^	13.27 ^a^	17.58 ^b^	17.50 ^b^	0.0001	0.0001	0.0602
Cold carcass weight, kg	12.96 ^a^	17.52 ^b^	12.49 ^a^	16.74 ^b^	16.49 ^b^	0.0001	0.0001	0.1084
Carcass shrink, %	5.75	5.20	5.87	4.88	5.69	0.6426	0.7638	0.6476
Dressing percentage, %	43.26	43.62	41.07 ^a^	44.09 ^b^	45.17 ^b^	0.6660	0.0009	0.2467
Loin area, cm^2^	13.03 ^a^	15.61 ^b^	13.58	14.81	14.57	0.0183	0.5914	0.1427
Primal cut, kg								
Neck	0.86	1.03	0.84	1.04	0.95	0.0512	0.1841	0.6073
Shoulder	3.52 ^a^	4.92 ^b^	3.45 ^a^	4.50 ^b^	4.71 ^b^	0.0001	0.0001	0.0519
Fore shank	0.61 ^a^	0.83 ^b^	0.61 ^a^	0.79 ^b^	0.76 ^b^	0.0001	0.0034	0.1442
Breast	0.29 ^a^	0.38 ^b^	0.30 ^a^	0.37 ^b^	0.34 ^b^	0.0004	0.0379	0.7241
Rack	1.57 ^a^	2.08 ^b^	1.14 ^a^	2.11 ^b^	1.96 ^b^	0.0001	0.0001	0.0302
Loin	0.95 ^a^	1.34 ^b^	0.94 ^a^	1.23 ^b^	1.27 ^b^	0.0001	0.0001	0.4599
Flank	0.48 ^a^	0.66 ^b^	0.43 ^a^	0.66 ^b^	0.63 ^b^	0.0001	0.0001	0.7334
Leg	4.54 ^a^	6.10 ^b^	4.49 ^a^	5.81 ^b^	5.67 ^b^	0.0001	0.0001	0.2562
Hind shank	0.56 ^a^	0.73 ^b^	0.54 ^a^	0.69 ^b^	0.70 ^b^	0.0001	0.0001	0.3631

^a,b^ Least squares means in same row within treatment differ significantly at *p* < 0.05.

**Table 3 animals-14-01217-t003:** Quality characteristics of loin chops from purebred and crossbred (wool × hair) hair sheep lambs grazed on tall fescue pastures with or without supplementation (soy hull, SH; corn gluten feed, CGF).

	Treatment					*p*-Value		
	Breed Type, B		Diet, D			Main		Interaction
Parameter	Purebred	Crossbred	Pasture Only	Pasture + CGF	Pasture + SH	B	D	B × D
Fresh								
L* value, lightness	36.93	37.21	37.89 ^a^	36.84 ^b^	36.48 ^b^	0.3296	0.0003	0.0001
a* value, redness	12.77 ^a^	11.64 ^b^	12.15	12.44	12.04	0.0001	0.1194	0.0057
b* value, yellowness	11.13 ^a^	10.33 ^b^	11.25 ^a^	10.75 ^b^	10.68 ^b^	0.0023	0.0057	0.0001
Myoglobin (Mb), mg/g	8.04	7.35	6.95	7.83	8.30	0.5234	0.6075	0.08776
Metmyoglobin MetMb), %	34.84	33.51	31.62	38.46	32.45	0.6340	0.0939	0.0001
TBARS, mg/kg	0.67	0.76	0.80 ^a^	0.81 ^a^	0.54 ^b^	0.3071	0.0264	0.6002
Ultimate pH	5.64	5.66	5.64	5.67	5.64	0.7820	0.8301	0.3941
Cooked								
Cooking loss, %	16.51	17.86	17.94	16.70	16.91	0.2686	0.6679	0.2048
WBSF, kg/cm^3^	3.00 ^a^	3.80 ^b^	3.46 ^a^	3.13 ^b^	3.61 ^a^	0.0001	0.0398	0.0080

^a,b^ Least squares means in same row within treatment differ significantly at *p* < 0.05.

**Table 4 animals-14-01217-t004:** Proximate and fatty acid composition of longissimus muscle (LM; intramuscular fat) from purebred and crossbred (wool × hair) hair sheep lambs grazed on tall fescue pastures with or without supplementation (soy hull; SH or corn gluten feed; CGF).

Item	Treatment					*p*-Value		
	Breed Type, B		Diet, D			Main		Interaction
	Purebred	Crossbred	Pasture Only	Pasture + CGF	Pasture + SH	B	D	B × D
Proximate Composition, %								
Moisture	75.21	75.29	76.07 ^a^	74.73 ^b^	74.94 ^b^	0.7518	0.0001	0.1538
Crude protein	20.93 ^a^	21.51 ^b^	21.11	21.41	21.15	0.0087	0.4849	0.9890
Ether extracted, fat	1.48	1.50	0.78 ^a^	1.78 ^a^	1.92 ^b^	0.8691	0.0001	0.2339
Ash	1.34	1.28	1.39 ^a^	1.29 ^ab^	1.24 ^b^	0.0784	0.0014	0.7554
Fatty acid, %								
C10:0	0.60	0.44	0.97 ^a^	0.29 ^b^	0.30 ^b^	0.0808	0.0001	0.7089
C12:0	0.23 ^a^	0.28 ^b^	0.34 ^a^	0.20 ^b^	0.22 ^b^	0.0015	0.0001	0.0004
C14:0	1.42	1.30	1.42	1.34	1.32	0.4422	0.8694	0.5335
C14:1n5	0.43	0.48	0.56 ^a^	0.38 ^b^	0.43 ^b^	0.0968	0.0001	0.3962
C16:0, *iso*	0.58	0.61	0.54 ^a^	0.59 ^a^	0.66 ^b^	0.1206	0.0003	0.4698
C16:0	16.44 ^a^	17.48 ^b^	15.86 ^a^	18.07 ^b^	16.95 ^ab^	0.0131	0.0004	0.5357
C16:1n7, *trans*	0.95	0.98	1.02 ^a^	0.87 ^b^	1.00 ^a^	0.3334	0.0005	0.4836
C16:1n7	2.20	2.00	3.20 ^a^	1.59 ^b^	1.51 ^b^	0.3336	0.0001	0.8384
C17:0	1.32	1.36	1.47 ^a^	1.20 ^b^	1.36 ^a^	0.3245	0.0001	0.8520
C17:1n7	0.57	0.59	0.66 ^a^	0.50 ^b^	0.57 ^b^	0.1602	0.0001	0.4152
C18:0	21.3	21.7	21.2	21.79	21.60	0.4786	0.6569	0.3306
C18:1n9, *trans*	0.40	0.41	0.42	0.39	0.40	0.4252	0.6881	0.4957
C18:1n11, *trans*	1.10	1.19	1.37 ^a^	1.04 ^b^	1.02 ^b^	0.1142	0.0001	0.4882
C18:1n9	31.25	32.56	29.10 ^a^	32.79 ^b^	33.82 ^b^	0.1722	0.0006	0.5788
C18:2n6, *trans*	0.26	0.28	0.40 ^a^	0.17 ^c^	0.24 ^b^	0.1166	0.0001	0.0420
C18:2n6	7.10	6.67	7.37	7.16	6.13	0.4839	0.2214	0.8565
C18:2, *c9,t11*	0.45	0.41	0.46	0.42	0.40	0.3759	0.6464	0.8367
C18:3n3	1.21	1.08	1.72 ^a^	0.76 ^b^	0.95 ^b^	0.2469	0.0001	0.1700
C18:3n6	0.53	0.58	0.50 ^a^	0.52 ^a^	0.64 ^b^	0.1574	0.0058	0.7175
C20:1n9	0.41	0.36	0.60 ^a^	0.27 ^b^	0.28 ^b^	0.3493	0.0006	0.5785
C20:2n6	0.38	0.37	0.51 ^a^	0.33 ^b^	0.29 ^b^	0.8896	0.0066	0.6442
C20:4n6	2.84	2.15	3.31 ^a^	2.10 ^b^	2.06 ^b^	0.0822	0.0184	0.4069
C20:5n3	0.71	0.54	1.06 ^a^	0.36 ^b^	0.46 ^b^	0.1943	0.0002	0.5808
C22:5n3	0.87	0.69	1.27 ^a^	0.52 ^b^	0.55 ^b^	0.2248	0.0007	0.3578
∑SFA ^d^	41.89	43.17	41.80	43.48	42.41	0.2088	0.2181	0.4901
∑MUFA ^e^	37.31	38.57	36.93	37.83	39.03	0.2481	0.0863	0.5343
∑PUFA ^f^	14.35	12.77	16.60	12.34	11.72	0.3080	0.0999	0.5125
∑n-3 ^g^	2.79	2.31	4.05 ^a^	1.64 ^b^	1.96 ^b^	0.2220	0.0004	0.3660
∑n-6 ^h^	11.56	10.46	12.55	10.70	9.76	0.3510	0.1498	0.5840
∑n-6/n-3	4.14	4.53	3.10	6.52	4.97	0.2865	0.0751	0.4768

^a,b,c^ Least squares means in same row within treatment differ significantly at *p* < 0.05. ^d^ SFA = saturated fatty acids; C10:0–C18:0. ^e^ MUFA = monounsaturated fatty acids; C14:1n5–C20:1n9. ^f^ PUFA = polyunsaturated fatty acids; C18:2n6, trans-C22:5n3. ^g^ n-3 = n-3 PUFA; C18:3n3, C20:5n3 and C22:5n3. ^h^ n-6 = n-6 PUFA; C18:2n6, trans, C18:2n6, C18:2, c9,t11, C18:3n6, C20:2n6 and C20:4n6.

**Table 5 animals-14-01217-t005:** Fatty acid composition and weight percent of fatty acid methyl esters in the subcutaneous fat of purebred and crossbred (wool × hair) hair sheep lambs grazed on tall fescue pastures with or without supplementation (soy hull; SH or corn gluten feed; CGF).

Fatty Acid, %	Treatment					*p*-Value		
	Breed Type, B		Diet, D			Main		Interaction
	Purebred	Crossbred	Pasture Only	Pasture + CGF	Pasture + SH	B	D	B × D
C10:0	0.24	0.23	0.27	0.23	0.21	0.7630	0.1401	0.9981
C12:0	0.16	0.13	0.12	0.16	0.15	0.2156	0.4918	0.8632
C14:0	2.43	2.39	2.65	2.32	2.25	0.8154	0.1587	0.4383
C14:1n5	0.88	0.91	0.87	0.87	0.94	0.4786	0.3013	0.9748
C16:0, *iso*	0.40	0.46	0.44 ^ab^	0.35 ^a^	0.49 ^b^	0.1131	0.0169	0.9059
C16:0	18.22	18.94	17.61 ^a^	19.62 ^b^	18.50 ^ab^	0.2270	0.0331	0.3693
C16:1n7, *trans*	0.47 ^a^	0.54 ^a^	0.54	0.46	0.51	0.0496	0.1829	0.6436
C16:1n7	1.40	1.23	1.32	1.41	1.22	0.0870	0.2640	0.0754
C17:0	2.25	2.13	1.95 ^a^	2.12 ^ab^	2.51 ^b^	0.3866	0.0099	0.6467
C17:1n7	1.58	1.34	1.10 ^a^	1.57 ^ab^	1.71 ^b^	0.1241	0.0108	0.6216
C18:0	15.43	16.00	20.30 ^a^	13.90 ^b^	13.00 ^b^	0.7898	0.0224	0.1306
C18:1n9, *trans*	0.36	0.38	0.44	0.33	0.35	0.7266	0.4332	0.8231
C18:1n11, *trans*	0.39	0.45	0.57 ^a^	0.26 ^c^	0.43 ^b^	0.1255	0.0001	0.2527
C18:1n9	46.45	44.84	43.32	46.51	47.09	0.3160	0.1471	0.1328
C18:2n6, *trans*	0.64	0.70	0.66	0.66	0.69	0.1191	0.6814	0.9622
C18:2n6	2.00	2.32	1.57 ^a^	2.59 ^b^	2.32 ^b^	0.0594	0.0001	0.9150
C18:2, *c9,t11*	0.20	0.21	0.29 ^a^	0.18 ^b^	0.16 ^b^	0.6176	0.0043	0.3783
C18:3n3	1.16	1.27	0.99 ^a^	1.15 ^a^	1.51 ^b^	0.2828	0.0004	0.9222
C18:3n6	0.75	0.84	0.88 ^a^	0.69 ^b^	0.83 ^a^	0.1333	0.0427	0.9106
C20:1n9	0.08	0.08	0.08	0.08	0.08	0.4684	0.7877	0.9766
C20:2n6	0.08	0.10	0.11	0.09	0.08	0.1128	0.0716	0.4754
C20:4n6	0.15	0.15	0.14	0.15	0.17	0.9051	0.3395	0.4574
C20:5n3	0.05	0.06	0.08 ^a^	0.04 ^b^	0.05 ^a^	0.8076	0.0013	0.3570
C22:5n3	0.17	0.18	0.19 ^a^	0.14 ^b^	0.18 ^ab^	0.5189	0.0158	0.4570
∑SFA ^c^	39.13	40.28	43.34	38.70	37.11	0.4729	0.1247	0.6217
∑MUFA ^d^	51.61	49.77	48.24	51.49	52.33	0.2970	0.2659	0.5626
∑PUFA ^e^	5.20	5.83	4.91	5.69	5.99	0.3952	0.1286	0.6483
∑n-3 ^f^	1.38	1.51	1.26 b	1.33 b	1.74 a	0.5364	0.0058	0.5787
∑n-6 ^g^	3.82	4.32	3.65	4.36	4.25	0.3246	0.1899	0.6832
∑n-6/n-3	2.77	2.87	2.90	3.28	2.44	0.4305	0.0979	0.5787

^a,b^ Least squares means in same row within treatment differ significantly at *p* < 0.05. ^c^ SFA = saturated fatty acids; C10:0–C18:0. ^d^ MUFA = monounsaturated fatty acids; C14:1n5–C20:1n9. ^e^ PUFA = polyunsaturated fatty acids; C18:2n6, trans-C22:5n3. ^f^ n-3 = n-3 PUFA; C18:3n3, C20:5n3 and C22:5n3. ^g^ n-6 = n-6 PUFA; C18:2n6, trans, C18:2n6, C18:2, c9,t11, C18:3n6, C20:2n6 and C20:4n6.

## Data Availability

The data presented in this study are available on request from the corresponding author.

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
