# Peer review of "Impact of Agro-Byproduct Supplementation on Carcass Traits and Meat Quality of Hair Sheep and Wool × Hair Crossbreds Grazing on Fescue Pasture"

_animals, 2024, doi:10.3390/ani14081217_

Round 1

Reviewer 1 Report

Comments and Suggestions for Authors

the text contains many inaccuracies and/or careless errors. I have highlighted many areas where English expression could be improved.

In summary, paper will need of major revision.

Simple Summary and Abstract: out of words, more than the specified number of words

L61 By adding published test results, how much does the growth performance of a sheep breed increase with a sheep breed?

Materials and Methods, In order to facilitate the reader to read, it should be divided into more detailed writing, such as experimental design. . Statistical analysis

L165: change (P < 0.0132) to (P < 0.05)

L17-19:rewriteThe study revealed that supplementation played a more crucial role than crossbreeding in enhancing carcass quality under the conditions of this experiment.”。

L45:change[3,2,4]to [2-4]。

L184:"the feeding regime" refers to the feeding regime

L223:rewrite ----The CIE L*, a* and b* values of lamb chops were affected (P < 0.006)

by supplementation, breed type, and breed type and supplementation, respectively.

L342:“corn gluten feedtoCGF”。

L68: soy hull to soy hull (SH)

4.:rewrite Conclusion. The reader should be told what conclusions you draw from the experiment, whether it is too much to implement, whether it is better to hybridize, or to supplement something

Author Response

Reviewer 1

The text contains many inaccuracies and/or careless errors. I have highlighted many areas where English expression could be improved.

In summary, paper will need of major revision.

Simple Summary and Abstract: out of words, more than the specified number of words

Simple Summary and Abstract were within the 200-word limit (198 words each), but an effort was made to reduce the length (remove redundant wording and remove details on materials and methods).

L61 By adding published test results, how much does the growth performance of a sheep breed increase with a sheep breed?

General values for growth rates and carcass weights have been added.

Materials and Methods, In order to facilitate the reader to read, it should be divided into more detailed writing, such as experimental design. . Statistical analysis

Subheadings have been added.

L165: change (P < 0.0132) to (P < 0.05)

Correction made

L17-19:rewrite“The study revealed that supplementation played a more crucial role than crossbreeding in enhancing carcass quality under the conditions of this experiment.”。

Sentence was re-written.

L45:change“[3,2,4]” to [2-4]。

Correction was made.

L184:"the feeding regime" refers to the feeding regime

Not clear as to the requested change

L223:rewrite ----The CIE L*, a* and b* values of lamb chops were affected (P < 0.006)

by supplementation, breed type, and breed type and supplementation, respectively.

Sentence was revised as follows: The CIE L*, a* and b* values of lamb chops were affected (P < 0.006)

by supplementation, breed type, and the interaction of breed type and supplementation.

L342:“corn gluten feed”to“CGF”。

Correction was made

L68: soy hull to soy hull (SH)

Correction was made

4.:rewrite Conclusion. The reader should be told what conclusions you draw from the experiment, whether it is too much to implement, whether it is better to hybridize, or to supplement something

The conclusion section was greatly reduced and more specifically aligned with the findings of the experiment.

Reviewer 2 Report

Comments and Suggestions for Authors

Comments and Suggestions for Authors

Manuscript ID: animals-2926729-peer-review-v1

entitled " Impact of Supplementing Agro-byproduct on Carcass Trait and Meat Quality of Hair Sheep and Wool x Hair crossbreds Grazing on Fescue Pasture."

This study suggests that crossbreeding aligns with market demands, but achieving optimal carcass weight and quality may require nutritional interventions such as dietary supplements like soy hull and corn gluten feed, which are cost-effective alternatives to traditional grain-based supplements.

A study aims to evaluate the effects of crossbreeding and dietary supplementation on carcass traits and meat quality of hair sheep and their crossbreds grazing on fescue pastures, aiming to optimize nutritional strategies using agro-byproducts.

This work needs minor revision. It could be considered for publication after a minor revision.

General comments.

1.     The abstract should show the detailed results of the study.

2.     In the tables, change significant letters of breed type effect into capital letters (A and B) to differ from the significant letters of diet effect.

3.     3. L189-191 delete these lines because they are repeated.

4.     L172-175 transfer into after L188. And delete the weight of primal cuts because it mentioned again in later in line 206.

5.     L222 remove (P < 0.006).

6.     L374 correct ‘where dietary supplements contributed to increased fat and ash levels’ to where dietary supplements contributed to increased fat level.

7.     L415-422 summarize these results.

8.     L426 please check the sentence ‘that these dietary saturated fatty acids’

9.     L445 and 450 change the LM to the subcutaneous fat.

10. L452, 453 delete this interpretation.

Author Response

Reviewer 2

Manuscript ID: animals-2926729-peer-review-v1

entitled " Impact of Supplementing Agro-byproduct on Carcass Trait and Meat Quality of Hair Sheep and Wool x Hair crossbreds Grazing on Fescue Pasture."

This study suggests that crossbreeding aligns with market demands, but achieving optimal carcass weight and quality may require nutritional interventions such as dietary supplements like soy hull and corn gluten feed, which are cost-effective alternatives to traditional grain-based supplements.

A study aims to evaluate the effects of crossbreeding and dietary supplementation on carcass traits and meat quality of hair sheep and their crossbreds grazing on fescue pastures, aiming to optimize nutritional strategies using agro-byproducts.

This work needs minor revision. It could be considered for publication after a minor revision.

General comments.

  1. The abstract should show the detailed results of the study.

More detailed results of major findings along with the significance level were included in the abstract.

  1. In the tables, change significant letters of breed type effect into capital letters (A and B) to differ from the significant letters of diet effect.

The format and footnote of Tables 2, 3, 4, and 5 were changed to more accurately reflect the comparisons that were made. The differences noted by the superscripts referred to differences within treatment, so the capital letters should not be needed.

  1. 3. L189-191 delete these lines because they are repeated.

Sentence was deleted and reference to Table 2 was incorporated into next sentence

  1. L172-175 transfer into after L188. And delete the weight of primal cuts because it mentioned again in later in line 206.

The sentenced was moved to lead of the following paragraph and reference to primal cuts was deleted.

  1. L222 remove (P < 0.006).

‘(P<0.006)’ was removed

  1. L374 correct ‘where dietary supplements contributed to increased fat and ash levels’ to where dietary supplements contributed to increased fat level.

Sentence was modified as requested

  1. L415-422 summarize these results.

The description of the interaction was reduced in length.

  1. L426 please check the sentence ‘that these dietary saturated fatty acids’

Sentence was corrected to state:’ Palmitic acid was significantly lower in purebred and pasture-only lambs (Table 4)’.

  1. L445 and 450 change the LM to the subcutaneous fat.

 Corrections were made

  1. L452, 453 delete this interpretation.

Interpretation was deleted

Reviewer 3 Report

Comments and Suggestions for Authors

General comments:

This study mainly aimed to determine the effect of crossbreeding and diet supplementation with two types of agro-byproducts on carcass yield and composition from lambs. This is relevant applied research which fit the scope of the journal. The introduction section is well done regarding the background and justification of this experiment. In M&M some additional information should be incorporated regarding the sample size of each group and the statistical analysis. The values of the results are mainly described in four tables. These tables describe significant differences within each group. This procedure can be enough for traits with significant B x D interaction. However, in some traits this interaction is significant. This mean that at least one of the six groups present a different pattern regarding the influence of breed and diets. In these situations, it is important to report what groups present better results and why. They allow to made a deep discussion, and extract a more specific conclusion. The readers need or wish to know hat specific groups is more adequate for rearing purpose. In the present version, the conclusion is too general: the crossbreeding lambs and diets with supplementation of agro-byproducts present better results. The discussion is adequate regarding the (overral) presentation of the results and supported by literature, but should be improved regarding the previous comments. Probably, a separation between results and discussion fit better, if the authors choose to exploit the data analysis. The abstract should report the main findings and conclude what supplement can add (or not) the better value for carcass yield a composition. I believe that a moderate revision can solve all these points.  

Specific comments:

L81-95: There are six groups (two genotypes/breeds and three diets). Please report the sample size (number of animals) for each group (n=6?). The initial body weight of each genotype in each group was similar for all groups?

L144-151:  I suggest to add an equation for this GLM including the two-Way Interactions. Also, a post hoc test was not reported to evaluate differences between pairs (but it was used).

L165: “…249.60 ± 9.516 g, respectively) …”. Please check the manuscript for this issue.

L170-171, 187-188: This aspect was reported by the significant BxD interaction on ADG? I.e., the response of crossbreeding lambs is better than pure lines when supplemented by agro-byproducts.

L164: “…249.60 ± 9.516 g) …”. Please be consistent in the presentation of values. Please check the manuscript for this issue

L176: The presentation and readability of the tables fit well when you start the row with “a” as superscript letter (moreover purebred serve as “control group”).

L179: Please add as caption the definition for superscript letter (same row within breeds or diets).

L201-203: There is any justification for this occurrence?

L224-225: We note that the authors report interactions but not the groups causing them. It is important to know this pattern to conclude what group show better results. I suggest to pay attention for this issue in the manuscript.

L474-488: In my opinion, this conclusion is too general. I think that you have results to support more specific and applied conclusion, namely to determine what group(s) express the better results regarding the studied traits.

Author Response

Reviewer 3

The authors appreciate the constructive comments by the reviewer and attempted to address all points raised in a major revision of the manuscript.  Revisions were made using tracking change in MS Word and should be readily apparent.

Specific comments:

L81-95: There are six groups (two genotypes/breeds and three diets). Please report the sample size (number of animals) for each group (n=6?). The initial body weight of each genotype in each group was similar for all groups?

It has been incorporated in the revised manuscript. Sentence was re-written.

L144-151:  I suggest to add an equation for this GLM including the two-Way Interactions. Also, a post hoc test was not reported to evaluate differences between pairs (but it was used).

It has been used for Mixed Model.

L165: “…249.60 ± 9.516 g, respectively) …”. Please check the manuscript for this issue.

Correction  made.

L170-171, 187-188: This aspect was reported by the significant BxD interaction on ADG? I.e., the response of crossbreeding lambs is better than pure lines when supplemented by agro-byproducts.

Sentence was rewritten

L164: “…249.60 ± 9.516 g) …”. Please be consistent in the presentation of values. Please check the manuscript for this issue

Correction  made.

L176: The presentation and readability of the tables fit well when you start the row with “a” as superscript letter (moreover purebred serve as “control group”).

Not clear as to requested change.

L179: Please add as caption the definition for superscript letter (same row within breeds or diets).

Correction made.

L201-203: There is any justification for this occurrence?

Sentence was re-written.

L224-225: We note that the authors report interactions but not the groups causing them. It is important to know this pattern to conclude what group show better results. I suggest to pay attention for this issue in the manuscript.

Sentence was re-written

L474-488: In my opinion, this conclusion is too general. I think that you have results to support more specific and applied conclusion, namely to determine what group(s) express the better results regarding the studied traits.

The conclusion section was greatly reduced and more specifically aligned with the findings of the experiment.

Reviewer 4 Report

Comments and Suggestions for Authors

The analytical testing done in this study is extensive, but I have key questions about the methodology.

There were 4 genotypes tested: BB, STC, DTxBB and DTxSTC. For the study to be completely randomized the DT group is missing. This is the first aspect.

Why don't the authors report the results for each genotype separately, but "toss" everything into one subgroup (purebreed vs. crossbreed)?

Which generation of hybrids is this? How were the individual breeds crossbred? 

The most important question for me is this: 

There is no information about what number of lambs of each genotype were fed each diet. How were the lambs divided? According to my calculations: There were 36 individuals in total, representing 4 different genotypes, and further divided into 3 diet groups, so even if the proportion of genotypes was equal in each nutrition group it comes out to 3 individuals per nutritional group. How were the statistics counted concerning this? What is the power of statistical tests with such a small sample size?

These questions raise the issue with the statistical analysis..

Minor comments:

M&M should be divided into subsections. The information about the number of animals in each subgroup should be added precisely. 

What was the nutritional value of the diet? Nutritional norms?

Table 1 Please add energy content.

Tables regarding the fatty acid profile of experimental additive and meat composition - please add the information about the share of SFA, MUFA, PUFA, including n-3 and n-6 FA and their ratio. 

Conclusion - should be changed regarding the title of the work, in this form it is too broad, too wordy, and does not point out the best interaction of genotype and diet

Author Response

Reviewer 4

The analytical testing done in this study is extensive, but I have key questions about the methodology.

There were 4 genotypes tested: BB, STC, DTxBB and DTxSTC. For the study to be completely randomized the DT group is missing. This is the first aspect.

Why don't the authors report the results for each genotype separately, but "toss" everything into one subgroup (purebreed vs. crossbreed)?

Which generation of hybrids is this? How were the individual breeds crossbred? 

The project was designed as a comparison of purebred landrace breed lambs (represented by both Barbados Blackbelly and St. Croix) with terminal sire lambs produced by mating these breeds to a Dorset rams, a medium size wool breed with some out-of-season breeding ability as the lambs were produced under an accelerated mating system. The intention was to quantify any benefits derived from use of a terminal sire to produce lambs for market. The production characteristics of the two breeds were considered sufficiently similar to be considered ‘landrace hair sheep’ in the context of this study. Care was taken to have two hair sheep breeds distributed equally each treatment groups.  The Materials and Methods were revised to more clearly describe this experimental design.

The most important question for me is this: 

There is no information about what number of lambs of each genotype were fed each diet. How were the lambs divided? According to my calculations: There were 36 individuals in total, representing 4 different genotypes, and further divided into 3 diet groups, so even if the proportion of genotypes was equal in each nutrition group it comes out to 3 individuals per nutritional group. How were the statistics counted concerning this? What is the power of statistical tests with such a small sample size?

These questions raise the issue with the statistical analysis..

As indicated in response to the earlier comment, Barbados Blackbelly and St. Croix lambs were considered ‘landrace sheep’ and compared as purebred and terminal sire lambs as the two breed types in this study. With three dietary treatments the experiment was analyzed as a 2 (breed types) x 3 (diets) design with 6 animals representing each cell.  The Materials & Methods were revised to more clearly describe this experimental design.

Minor comments:

M&M should be divided into subsections. The information about the number of animals in each subgroup should be added precisely. 

Subsections were added as requested

What was the nutritional value of the diet? Nutritional norms?

Table 1 Please add energy content.

Nutritional values of both forage and supplements are summarized in Table 1. Values of the net energy analysis of the diet components have been added.  No effort was made to estimate the extend the three diets may, or may not, have addressed the nutritional requirements of growing lambs as total intake was not measured in this study.

Tables regarding the fatty acid profile of experimental additive and meat composition - please add the information about the share of SFA, MUFA, PUFA, including n-3 and n-6 FA and their ratio. 

Information has been added to tables.

Conclusion - should be changed regarding the title of the work, in this form it is too broad, too wordy, and does not point out the best interaction of genotype and diet

The ‘Conclusion’ has been revised, and considerably shortened.  It has been more strictly focused on the findings of the study.

Round 2

Reviewer 1 Report

Comments and Suggestions for Authors

None

Reviewer 3 Report

Comments and Suggestions for Authors

This reviewer thanks the authors for providing this new version which readability was improved.

Just a note: it is recomended that the superscript letters should be inserted in each line of  Table 2 by alphabetic order, similar to Table 3 (please , also correct the last line of this Table)

Reviewer 4 Report

Comments and Suggestions for Authors

Dear Authors,

The information added made the manuscript clearer and easier to follow. 

I have found some small mistakes in the text like in line 248: "[32].ALoin chops", so please check it carefully once again.

Comments on the Quality of English Language

Quality of English language is correct.